# Multimodal Imaging of Osteosarcoma: From First Diagnosis to Radiomics

**DOI:** 10.3390/cancers17040599

**Published:** 2025-02-10

**Authors:** Maurizio Cè, Michaela Cellina, Thirapapha Ueanukul, Gianpaolo Carrafiello, Rawee Manatrakul, Phatthawit Tangkittithaworn, Suphaneewan Jaovisidha, Praman Fuangfa, Donald Resnick

**Affiliations:** 1Postgraduation School in Radiodiagnostics, Università degli Studi di Milano, Via Festa del Perdono 7, 20122 Milan, Italy; maurizio.ce@unimi.it (M.C.); gianpaolo.carrafiello@unimi.it (G.C.); 2Radiology Department, ASST Fatebenefratelli Sacco, Piazza Principessa Clotilde 3, 20121 Milan, Italy; michaela.cellina@asst-fbf-sacco.it; 3Department of Diagnostic and Therapeutic Radiology, Faculty of Medicine Ramathibodi Hospital, Mahidol University, Bangkok 10400, Thailand; thirapapha.uea@student.mahidol.ac.th (T.U.); rawee.man@mahidol.ac.th (R.M.); phatthawit.tan@mahidol.ac.th (P.T.); suphaneewan.jao@mahidol.ac.th (S.J.); 4Radiology Department, Fondazione IRCCS Cà Granda Ospedale Maggiore Policlinico, Via Francesco Sforza 35, 20122 Milan, Italy; 5Department of Radiology, University of California, San Diego, CA 92093, USA

**Keywords:** osteosarcoma, primary malignant bone tumor, bone cancer, Bone-RADS, multimodal imaging, artificial intelligence

## Abstract

Osteosarcoma is a primary malignant bone tumor that produces an osteoid matrix. Imaging is crucial for detection, characterization, and treatment planning, though it can be challenging, especially in the early stages. Conventional radiography is typically the first modality used to detect suspicious findings, while the Bone-RADS system helps describe imaging results and stratify risk. CT provides detailed insight into bone architecture and the osteoid matrix, while MRI is essential for assessing tumor spread to adjacent soft tissues, the medullary canal, joints, and neurovascular structures. The review includes clinical case examples and explores the role of artificial intelligence in improving osteosarcoma diagnosis.

## 1. Introduction

Osteosarcoma is a primary bone tumor characterized by mesenchymal cells producing an osteoid matrix [1,2]. It is the most common primary bone cancer not involving blood cells [1,2]. Although rare in the general population, with an incidence rate of approximately 4–5 cases per million people annually (less than 1% of all cancer cases) [1,3], large-scale epidemiological studies have revealed a rising incidence in recent decades [4]. Osteosarcoma has traditionally been considered a tumor with a bimodal incidence, showing a primary peak in children and adolescents and a secondary peak in the elderly over 80 years of age [1,4]. However, recent insights challenge the typical bimodal pattern, suggesting instead a single consistent peak of incidence in the second decade of life [5]. In children and adolescents, it represents the most common primary malignant mesenchymal tumor, accounting for approximately 80% of all malignant bone tumors and making up about 2.4% of all childhood cancers [1]. In individuals over fifty years of age, osteosarcoma accounts for about 50% of all primary malignant bone tumors, with a higher proportion of secondary osteosarcomas [2]. It is more common in males than females, with a male-to-female ratio of approximately 1.5:1 to 2:1 [4]. Although some studies suggest variations in incidence across different ethnic groups, with higher rates reported in Black and Hispanic populations, global data remain inconsistent [4]. Risk factors for this tumor include prior therapeutic radiation exposure, such as from cancer treatment, tall stature, high birth weight, and conditions like Paget’s disease or hereditary multiple exostoses, which involve rapid bone growth or turnover and can also increase the risk of developing osteosarcoma [4,6]. It has been linked to several cancer predisposition syndromes, including hereditary retinoblastoma (associated with RB1 gene mutations) [7,8], Li-Fraumeni syndrome (related to TP53 gene mutations) [9,10], and pathogenetic germline variants in individuals with osteosarcoma, especially in the young [11].

Osteosarcoma most commonly arises near the metaphysis of long bones, particularly around the knee joint [12]. The femur is the most frequently affected site, accounting for about 42–46% of cases, followed by the tibia (about 19–24%) and the humerus (about 10–12%). In older adults, osteosarcoma can also occur in other bones, such as the pelvis, ribs, and jaw [2]. The World Health Organization’s (WHO) classification of soft tissue and bone tumors recognizes various types of osteosarcomas with differing levels of malignancy [13]. Clinically, osteosarcoma typically presents with deep-seated progressively worsening pain [14]. Other symptoms may include a palpable mass and restricted movement and, in 5–10% of cases, the tumor may lead to a pathological fracture.

Imaging plays a crucial role in the detection, diagnosis, and staging of osteosarcoma [12,15,16]. Although these tumors typically show aggressive imaging features, the initial diagnosis can be challenging and subtle, particularly in the early stages, in which they may be confused with benign mimickers or lesions with low malignant potential [14,15]. Delays in diagnosis can occur because symptoms such as localized pain or swelling are often attributed to benign conditions, including sports injuries or growing pains in younger patients [16]. While conventional radiography is often the first imaging modality applied to the analysis of bone tumors such as osteosarcomas, the integration of computed tomography (CT) and magnetic resonance imaging (MRI) is essential for an accurate characterization of the lesions [17]. CT scans provide a more comprehensive evaluation of bone structures compared to conventional radiography, whereas MRI is particularly valuable for assessing the tumor’s extension into surrounding soft tissues [18]. Additionally, bone scan and PET-CT play an important role in staging and therapeutic planning [19].

The diverse presentation and aggressive nature of osteosarcoma necessitate precise imaging for both the initial diagnosis and monitoring of treatment response, given the tumor’s complex and variable characteristics. Therefore, integrating various imaging modalities is crucial for a thorough evaluation of the tumor and effective therapeutic planning (Table 1) [20]. In support of the visual assessment by radiologists, the intensive application of artificial intelligence to the analysis of biomedical images in the last decade has facilitated a paradigm shift in oncological imaging [21]. Radiomics plays a promising role by enabling the extraction of a wide range of quantitative features from imaging, which enhances tumor diagnosis, characterization, and prediction of therapeutic response [22,23]. However, despite technological and methodological advancements that have overcome some of the initial challenges in cancer imaging, particularly in lung, breast, and brain tumors [21,24,25,26], in the study of bone tumors, the clinical translation of these models was quite constrained [27]. For this reason, the role of the radiologist remains indispensable. Nevertheless, radiologists must be mindful of these new opportunities to contribute actively to ongoing healthcare advancements.

## 2. Imaging Assessment of Osteosarcoma

Conventional radiography is a non-invasive, low-cost, and widely accessible modality, and it is usually the first-line imaging for the diagnostic assessment of osteosarcomas and bone tumors in general [28,29]. The radiographic features of osteosarcomas are typical of aggressive bone tumors, and it is essential for radiologists to recognize these signs for the early diagnosis and appropriate management [12,29,30]. In approaching a bone lesion, several key factors such as the anatomical location and radiographic characteristics of the lesion, patient age, and clinical presentation must be carefully considered in the differential diagnosis and in planning further diagnostic and therapeutic decisions [20,31]. The density of the lesion on radiographs and CT can be categorized as radiolucent, sclerotic, or mixed [20,31,32]. A radiolucent lesion is characterized by lower attenuation compared to normal bone, while a sclerotic lesion is denser than the surrounding trabecular bone. In many osteosarcomas, the lesion contains a combination of osteoblastic (sclerotic) and osteolytic (lucent) areas (Figure 1) [12,30].

Matrix and tumor mineralization are important features that can be evaluated. The term “matrix” refers to the type of tissue present in the tumor (e.g., osteoid, chondral, fibrous), while “mineralization” pertains to the calcification of the matrix. Bone tumors most commonly exhibit a calcified osteoid matrix, which is easily recognizable [31]. However, some variants may also show cartilage components or a chondroid matrix, with its typical popcorn-like appearance on radiography or significant amounts of fibrous tissue [12,30].

In recent years, several diagnostic scoring systems have been introduced to further aid in classifying bone lesions and predict the risk of malignancy based on radiographic characteristics [33]. Among these, Bone-RADS is designed to provide a structured reporting system, similar to BI-RADS for breast imaging, which assigns a score to bone lesions based on their likelihood of malignancy, helping radiologists to stratify lesions and guide clinical management (Table 2 and Table 3) [32,34].

The numbers attributed to each feature are added to obtain a global Bone-RADS score (Table 3).

When evaluating osteosarcoma using the Bone-RADS score, several imaging features are crucial for assessing the tumor’s aggressiveness and extent. In osteosarcoma, the margins typically exhibit a board transition zone between the lesion and the surrounding perilesional space, presenting as ill-defined or with a moth-eaten appearance, reflecting the tumor’s aggressive nature (Figure 2).

In contrast, benign lesions typically present with well-defined or sclerotic margins [18]. Osteosarcomas grow rapidly, preventing the periosteum from having enough time to lay down a new layer of bone. As a result, Sharpey’s fibers stretch perpendicular to the bone, leading to the formation of new bone spicules that radiate outward. This creates a distinctive appearance known as the “sunburst” or “hair-on-end” pattern of periosteal reaction (Figure 3) [32,35,36]. It is frequently associated with osteosarcoma but can also occur with other aggressive bone lesions like Ewing’s sarcoma. Other features include the formation of Codman’s triangle (a periosteal reaction seen as the tumor elevates the periosteum).

The endosteal scalloping may show thinning or erosion of the inner bone surface, contributing to a permeative or moth-eaten appearance on imaging (Figure 4) [37].

The presence of fracture within an aggressive bone lesion is highly indicative of a malignant process (Figure 5).

Following the initial radiographic diagnosis, the assessment of osteosarcoma can benefit from a thorough evaluation using CT and MRI. These imaging modalities provide a more detailed assessment of the tumor’s composition, extent, and involvement of surrounding tissues, all of which are critical for formulating an effective treatment plan for osteosarcoma [38,39]. CT is the reference examination for the study of bone structures, providing a detailed visualization of regional bone anatomy [40], areas of bone resorption, and calcified bone matrix deposition while also improving the detection of pathological fractures. The impact of pathological fractures at diagnosis in patients with osteosarcomas is debated. [19]. In the EURAMOS-1 cohort, fractures were not associated with overall survival or event-free survival (EFS) in multivariable analysis [41]. However, in a study of 2847 patients from the Cooperative Osteosarcoma Study Group, Kelley et al. found significantly lower overall survival in adults with pathological fractures (HR = 1.89; *p* = 0.013), though no significant association was noted for EFS in adults or overall survival/EFS in children. By combining thin-slice CT reconstructions with different window parameters, CT enhances the depiction of various tissue components, including osseous and soft tissue tumor matrices, and produces highly detailed three-dimensional representations of the lesion and surrounding structures [42]. These renderings are particularly valuable in the assessment and surgical planning of osteosarcomas located in anatomically complex regions, such as the maxillofacial structures (Figure 6) and vertebrae (Figure 7) [42]. According to many authors, pathological fractures at diagnosis should not be considered an absolute indication for amputation, although they can lead to significant intraoperative bleeding, increased surgical complexity, and longer surgeries [43]. Initial management should include cast immobilization or external fixation to prevent tumor cell dissemination, followed by neoadjuvant chemotherapy and limb-salvage surgery. However, in cases of chemotherapy-resistant sarcomas, limb-salvage surgery is considered a relative contraindication [43,44].

Furthermore, CT is highly effective for guiding lesion biopsy due to its rapid execution and high spatial resolution. It ensures accurate placement of the biopsy needle, optimizing tissue sampling and minimizing complications by avoiding critical structures and targeting the most relevant areas of the tumor (Figure 8) [45].

CT is the preferred examination for comprehensive staging of osteosarcoma, as it enables the efficient detection of distant metastases [45], which are present in 10–15% of patients at diagnosis [46]. Metastases are most commonly found in the lungs (80%) [47], followed by the bones (30–35%) and lymph nodes (2%) [48]. In patients with a known diagnosis of osteosarcoma, lung nodule size is the primary factor for suspicion when larger than 5 mm [49,50]. Approximately 60% of osteosarcoma lung metastases are partially or fully calcified, providing an additional diagnostic clue [50] (Figure 9).

Furthermore, CT can also be used for local and distant complication assessment (Figure 10).

MRI is the preferred imaging modality for assessing tumor bulk, local disease extent, and its relationship with surrounding structures due to its superior tissue differentiation [19,38]. It provides a clear visualization of cartilage, muscles, fascia, and neurovascular bundles, and it represents the preferred modality to monitor tumor changes during treatments [51]. MRI protocols typically include both wide-field-of-view images, which allow for a comprehensive evaluation of the entire bone from joint to joint, and narrow-field-of-view images focused on the primary lesion for a better definition of lesion characteristics [38]. Imaging of the entire extremity is recommended, for instance, in cases of femoral osteosarcoma; this includes the hip, knee, and ankle joints to assess the longitudinal extent of the disease, evaluate both intraosseous and extraosseous involvement, and detect any skip lesions [48,52,53]. Skip metastases show a wide prevalence range (1% to 25%), likely due to differences in detection methods [54], and they were observed in about 16% of cases in a large cohort of 241 patients [55]. They are significantly associated with lung metastases, other skeletal metastases, and reduced survival rates but not with chemotherapy response [54,55]. It remains debated whether skip lesions result from local or systemic spread [55]. However, the detection of skip lesions is crucial, as they act as an independent variable associated with a very poor prognosis [56,57,58]. The smaller-field-of-view images include T1-weighted fat-suppressed and T2-weighted fat-suppressed sequences, as well as contrast-enhanced T1-weighted fat-suppressed images acquired in the axial plane and at least one longitudinal plane, depending on the localization of the disease. A 3–5 mm section thickness with a 0–20% intersection gap is usually applied [18]. Osteosarcoma typically presents on MRI as masses with heterogeneous low signal intensity on T1-weighted imaging (T1-WI), high signal intensity on T2-weighted imaging (T2-WI), and marked enhancement following gadolinium contrast administration (Figure 11) [19].

Depending on the histologic subtype and matrix predominance (osseous or chondroid), osteosarcomas may exhibit variable MRI signal characteristics. T1-weighted images are ideal for delineating the longitudinal extent of the tumor, its maximal distance from the nearest joint’s articular surface, and the intramedullary tumor component, which appears as an area of low signal intensity against the higher signal intensity of healthy bone marrow [59,60]. In contrast, fluid-sensitive sequences excel at characterizing extraosseous lesions and assessing the tumor’s relationship to the muscle compartment, neurovascular bundles, and adjacent joints [60]. However, they may overestimate the intramedullary extent due to their inability to distinguish the tumor from adjacent peritumoral reactive marrow edema and red marrow hyperplasia [61,62]. Necrotic regions display a variable signal on T1-weighted images, which can also appear hyperintense after fat suppression due to hemorrhaging, high signal intensity on T2-weighted images, and the absence of contrast enhancement. From a therapeutic planning perspective, epiphyseal extension is an important consideration, characterized by abnormal marrow signal intensity that extends continuously from the primary tumor into the epiphysis (Figure 12). This appears as hypointensity on T1-weighted images, hyperintensity on fluid-sensitive sequences, and increased post-contrast enhancement relative to skeletal muscle, suggesting direct tumor spread [63,64].

Joint invasion is another critical finding (Figure 13). Thick, linear, or nodular synovial enhancement, particularly in joints like the glenohumeral, indicates that the tumor has breached the joint capsule [38,65]. Recognizing joint invasion is essential for determining the appropriate surgical resection and ensuring that the joint is adequately addressed during surgery.

MRI is also the preferred method for assessing the invasion of adjacent soft tissues and the infiltration of critical structures, such as nearby organs and neurovascular bundles (Figure 14).

Overall, the accurate assessment of tumor extent and composition is essential not only for surgical planning but also for predicting therapeutic response. Recent evidence suggests that baseline MRI findings, as shown by Kanthawang et al., may help in predicting chemo-resistant osteosarcoma, with tumor size emerging as the strongest predictor [66]. In this recent study of 95 patients with newly diagnosed high-grade osteosarcoma, several associated factors were found to be with poor histologic response post-chemotherapy. These factors include a tumor volume greater than 150 mL, the longest diameter exceeding 7 cm, necrotic areas comprising over 50% of the tumor volume, intra-articular spread, and peritumoral soft tissue edema. Additionally, both the initial longest diameter and tumor volume were identified as independent predictors of histologic response [66]. Techniques such as diffusion-weighted imaging (DWI) and dynamic contrast-enhanced MRI (DCE-MRI) offer valuable insights into tumor biology, vascularity, and chemotherapy response. DWI detects variations in water molecule diffusion within tissues [67,68]. Since malignant tumors like osteosarcoma typically have densely packed cells, they restrict water molecule movements, resulting in high signal intensity on DWI and low ADC values; this characteristic helps distinguish osteosarcoma from less aggressive lesions or benign tumors [20]. Since bone tumors tend to become very heterogeneous as they grow, incorporating cystic-necrotic or hemorrhagic areas, restricted diffusion can assist in biopsy sampling along with contrast enhancement sequences (Figure 15).

Neoadjuvant chemotherapy significantly alters the composition of osteosarcoma tumors [69]. Currently, histopathological necrosis serves as the standard for evaluating chemotherapy response, but it can only be assessed after treatment completion [69]. For this reason, over the years, the role of imaging as a non-invasive surrogate has been explored to evaluate the degree of post-chemotherapy necrosis. The ADC value on DWI is a promising tool for monitoring the therapeutic response of primary bone sarcomas [19]. Hayashida et al. found that post-treatment changes in ADC values were significantly greater in tumors with ≥90% necrosis compared to those with less necrosis [70]. Recently, a study by Hao et al. on thirty-four osteosarcoma patients who received three courses of neoadjuvant chemotherapy showed that by week 9 post-chemotherapy, all patients had increased ADC values compared to pre-chemotherapy levels, indicating the chemotherapy’s impact on tumor cells. Furthermore, the responder group had a significantly higher difference in ADC (ΔADC) than the poor responder group, with increased necrosis reflecting a reduced restriction of water molecule motion and a better treatment response [71]. Other authors have found that the diffusion coefficient does not correlate with necrosis, but after adjusting for tumor volume, a significant correlation was observed, leading to the potential for a new parameter—diffusion per unit volume—that may offer more accurate insights [69]. Similarly to soft tissue sarcomas, the main drawback of DWI/ADC in bone tumors is image quality and lack of standardization [72].

Dynamic contrast-enhancement MRI provides valuable information for staging and treatment planning [20,65]. Enhanced T1-weighted images offer insights into tumor heterogeneity, revealing areas of cystic degeneration and necrosis. Additionally, these images can highlight the solid components within heterogeneous regions observed on T2-weighted images, aiding biopsy guidance and helping to detect joint involvement [20,65]. Although MRI is highly sensitive to joint invasion, careful analysis is essential to avoid false positives, which can lead to over-staging and unnecessarily radical surgical procedures [38]. When interpreting chemotherapeutic response, combining DWI and DCE-MRI with conventional MRI sequences (such as post-contrast T1-weighted, pre-contrast T1-weighted, and STIR) through machine learning approaches can reliably differentiate necrosis from viable tumors. A study by Teo et al. demonstrated excellent correlation between these imaging techniques and pathologist-estimated necrosis, as well as necrosis quantified from digitized histologic section images [73]. DCE-MRI analysis can be effectively enhanced by evaluating a range of quantitative or semi-quantitative parameters, such as the shape of the time–intensity curve, the area under the time–intensity curve, the wash-in rate, the influx volume transfer constant (Ktrans), the efflux rate constant (Kep), the relative extravascular extracellular space (Ve), and the relative vascular plasma space (Vp). These parameters are estimators of tumor neoangiogenesis [74]. Recently, Kalisvaart et al. used a machine learning approach to develop a model based on DCE-MRI features to predict the chemotherapy response in osteosarcoma patients. The optimal model found that a relative wash-in rate of <2.3 using whole slab segmentation best predicted poor histological response, achieving high accuracy and AUC scores. External validation confirmed consistent predictive performance, highlighting the wash-in rate as a reliable indicator for assessing chemotherapy response in osteosarcoma [75]. One important feature to assess is the presence of skip lesions, which are additional sites of tumors within the same bone but separated from the primary tumor by normal marrow. Identifying skip lesions is crucial for surgical planning, as they can significantly influence the extent of resection and overall treatment approach [48,52]. Muscle involvement is noted when the tumor directly penetrates surrounding muscle tissues, often encasing the myotendinous junction or causing displacement and signal intensity changes in imaging [76]. This can have a significant impact on surgical margins and postoperative function, as the tumor involvement of muscle requires careful consideration during resection. In more advanced cases, neurovascular involvement becomes a critical concern. This occurs when the tumor abuts, partially encases, or even infiltrates major vessels or nerves, sometimes leading to thrombosis of the neurovascular bundle. Such involvement not only increases the complexity and risk of surgery but also significantly affects treatment strategies [77].

The presence of lymphadenopathies, indicated by abnormal lymph node morphology or enlargement, can also suggest systemic spread. Pathological lymph nodes often display cortical thickening and a loss of the fatty hilum, a key sign of malignant involvement.

Bone scintigraphy (Figure 16), typically performed using technetium-99 m-labeled diphosphonates, is a valuable tool in the evaluation of osteosarcoma, particularly for detecting skeletal metastases and assessing the extent of disease [78].

Its high sensitivity allows for the identification of osseous abnormalities not readily apparent on conventional radiographs, making it essential in staging and monitoring disease progression. By visualizing areas of increased osteoblastic activity, the scan can distinguish between primary tumor involvement and distant metastatic sites. While bone scans are not specific for malignancy, their ability to provide whole-body imaging quickly and effectively makes them a critical adjunct in the diagnostic workup and follow-up of osteosarcoma patients.

The advent of newer modalities like PET/CT and MRI has gradually shifted some preference toward more advanced imaging, offering superior specificity and the simultaneous assessment of soft tissue involvement [79,80]. However, according to other studies, their performance is comparable [81]. The 18F-FDG PET/CT imaging modality combines metabolic and anatomical information by using a radioactive glucose analog that accumulates in cells with high glucose uptake, such as cancer cells, to assess tumor metabolic activity [82]. Studies have explored baseline and chemotherapy-induced changes in 18F-FDG PET/CT metrics, such as SUVmax, SUVmean, SUVpeak, the tumor-to-background ratio (TBR), total lesion glycolysis (TLG), and metabolic tumor volume (MTV) to evaluate histologic response in osteosarcomas [83]. Palmerini et al. found higher response rates in patients with a baseline SUVmax < 6 (64% vs. 20%, *p* = 0.05) [84]. Oh et al. highlighted cut-offs associated with favorable histologic response, including a post-chemotherapy SUVmax < 2–3, TBR > 0.46–0.60, and a 52–60% reduction in SUVmax [80]. Xu et al. also demonstrated that PERCIST criteria are more sensitive than RECIST v1.1 for detecting therapeutic responses [19,85].

Despite the challenges in diagnosing osteosarcoma due to nonspecific symptoms, radiological overlap with benign conditions, and the tumor’s complexity, multi-modal imaging approaches—combining X-ray (XR), MRI, CT, and PET—have significantly improved early detection and characterization. Additionally, AI-driven analysis enhances early detection by identifying subtle patterns that may be missed on conventional X-rays, particularly malignant features [86]. The complementary information provided by integrating MRI and CT allows for a more comprehensive evaluation of the tumor’s extent, heterogeneity, relationship with adjacent structures, and potential metastasis, including skip metastases, thus improving diagnosis and staging. Ongoing research into biomarkers and molecular imaging aims to offer non-invasive tools that further enhance the sensitivity and specificity of osteosarcoma detection [87].

## 3. Osteosarcoma Subtypes

The classification of osteosarcomas is notably complex [12,88]. Osteogenic tumors are classified based on several criteria, including radiographic, histopathological, and microscopic analyses [88]. Together, these diagnostic approaches reveal the multicellular complexity underlying the histological patterns of osteosarcoma. According to the WHO classification, osteosarcomas can be broadly divided into central and juxtacortical osteosarcoma (Table 4). Central osteosarcomas, originating in the bone’s intramedullary compartment, include conventional, telangiectatic, small-cell, and low-grade central variants [89]. Surface osteosarcomas arising near the bone surface, hence called juxtacortical or surface subtype, are classified into parosteal, periosteal, and high-grade surface subtypes [13,89,90]. However, this only provides a simplified view of the broader and more intricate landscape of osteosarcoma variants [91]. From a strictly radiological perspective, classification systems rely on several key features [20,28,31]. These include the tumor’s precise location within the bone (such as intramedullary or central, intracortical, surface, periosteal, or parosteal), the degree of cellular differentiation (high-grade or low-grade), and its histologic composition (including osteoblastic, chondroblastic, fibroblastic, fibrohistiocytic, telangiectatic, small-cell, or clear-cell subtypes) [12]. Additionally, the number of tumor foci (whether single or multicentric) and the condition of the underlying bone (whether normal or affected by conditions like Paget’s disease) are important factors in the classification. However, due to the rapid growth and highly aggressive nature of these tumors, which are often diagnosed in advanced stages, identifying the precise location of origin can be challenging, particularly in large lesions. This complexity adds a layer of difficulty in osteosarcoma diagnosis and management, requiring comprehensive radiological and histological evaluations to determine the appropriate classification and subsequent treatment approach.

### 3.1. Intramedullary (Central) Osteosarcoma

#### 3.1.1. Conventional Osteosarcoma

Conventional osteosarcoma is the most common form, accounting for 80% of cases, and it is a highly aggressive high-grade bone tumor [12,20,30]. It primarily affects individuals in their second decade of life, with 75% of cases occurring in those under the age of 25. The incidence is higher in males than in females, with a male-to-female ratio of 3:2 [13,30,89]. Approximately 90% of cases originate in the metaphysis of long bones, particularly in regions of rapid growth; initial involvement of the diaphysis occurs in about 10%, while epiphyseal involvement is rare [12]. However, evaluating epiphyseal involvement remains crucial, as metaphyseal cartilage does not effectively serve as a barrier [99]. In pediatric cases, extension into the epiphysis is observed in 75–88% of instances [100]. Histologically, the cells range from spindle-shaped to polyhedral, with nuclei that vary in appearance; mitotic cells are easily identifiable [53]. The tumor cells produce a matrix that can be osseous (“osteoblastic”), cartilaginous (“chondroblastic”), or fibrous (“fibroblastic”), though a combination of all three is often present [53,88]. The osteoid matrix of the tumor typically appears less dense than healthy bone and is characterized by an amorphous disorganized appearance (Figure 17).

On conventional radiography, they initially present as intramedullary lesions and are characterized by immature cloud-like bone formation and permeative eccentric osteolysis in metaphysis or metadiaphysis, with poorly defined non-sclerotic margins indicating aggressive behavior [12]. The rapid growth and destructive nature of the tumor prevent cortical expansion or the “blown out” appearance typically seen in other lesions [18]. The tumor’s density can vary from lytic to intensely sclerotic. The specific radiographic pattern varies depending on the tumor subtype, although there is a high overlap [53].

The osteoblastic subtype typically presents dense cloud-like mineralization and appears more sclerotic. In contrast, the chondroblastic subtype usually exhibits a more lytic appearance on radiographs, reflecting the presence of unmineralized tissue components. Similarly, the fibroblastic subtype also tends to show a lytic appearance due to the lack of a mineralized matrix. However, up to 90% of cases display a mixed lytic and sclerotic pattern (Figure 18).

Cortical destruction is common, with an extra-compartmental mass in the adjacent soft tissues in 90% of cases, where the osteoid matrix can often be visualized. Periosteal reactions are pronounced and can present as various forms. The presence of signs such as Codman’s triangle is common, while in the osteoblastic form, a robust periosteal reaction is more frequently observed, often displaying a “hair-on-end” or “sunburst” pattern.

On CT imaging, conventional osteosarcoma typically appears as a predominantly sclerotic lesion with frequent cortical bone invasion and disruption, along with exuberant osteoid matrix production. The dense mineralized osteoid matrix, often extending into the medullary cavity, is a defining feature easily identified on CT (Figure 17B,D). The periosteal reaction may appear as a “sunburst” pattern, with radiating spicules of new bone projecting into the soft tissue or as an elevated periosteum, equivalent to Codman’s triangle on the radiograph.

On MRI, depending on the histologic subtype and the predominance of an osseous or chondroid matrix, osteosarcomas may exhibit variable MRI signal characteristics. In areas with a bone-forming matrix, signal intensity is typically low across all sequences. Conversely, chondroid regions, common in periosteal, parosteal, and dedifferentiated osteosarcomas, present with low signal intensity on T1-WI, markedly high signal intensity on T2-WI, and contrast enhancement that is often peripheral, heterogeneous, and lobulated.

#### 3.1.2. Secondary Osteosarcoma

Secondary osteosarcoma is a particularly aggressive form of osteosarcoma that develops in bones already affected by underlying conditions predisposing to sarcomatous transformation. It is classified by the World Health Organization (WHO) under conventional osteosarcomas [13,101]. These conditions include Paget’s disease, poorly differentiated chondrosarcoma, and bones previously exposed to radiation. Secondary osteosarcoma typically arises in the fifth decade in individuals with Paget’s disease, around 11 years post-radiation therapy, or during the fifth to sixth decades when originating from chondrosarcoma degeneration. Compared to primary osteosarcoma, a considerable number of secondary osteosarcomas occurred at unfavorable sites [101]. The prognosis for secondary osteosarcoma is poor, with a rapid progression and a tendency for early metastasis. The diagnosis is primarily based on the patient’s clinical history and imaging findings of the underlying bone condition, as secondary osteosarcomas are histologically indistinguishable from conventional osteosarcomas.

In cases associated with Paget’s disease, secondary osteosarcoma often presents with a combination of lytic and thickened areas, which appear irregular [102,103]. There may also be cortical disruption and extra-osseous masses, reflecting the aggressive nature of the tumor within the pre-existing Pagetic bone. When secondary osteosarcoma develops in irradiated bones, particularly in pediatric patients, it can manifest as small lesions with mixed lytic and blastic changes [103]. The tumor may affect either the entire bone or specific segments, depending on the extent of prior radiation treatment [104,105,106].

Secondary osteosarcoma arising from chondrosarcoma typically shows cortical thickening, variable amounts of chondroid matrix, and cortical interruptions. It may also present with extra-osseous masses, highlighting its aggressive transformation from the pre-existing cartilaginous tumor.

#### 3.1.3. Teleangectatic Osteosarcoma

Telangiectatic osteosarcoma, also known as hemorrhagic osteosarcoma, is a relatively uncommon subtype, making up less than 10% of all osteosarcoma cases [92]. The retrospective study from Angelini et al. indicates a 10-year survival rate of approximately 60%, with most patients achieving a cure through a combination of neoadjuvant chemotherapy and limb-sparing surgery [93].

Telangiectatic osteosarcoma is considered a variant of conventional osteosarcoma. It primarily affects individuals in their second decade, with a male-to-female ratio of 2:1 [92]. Clinically, it is characterized by deep-seated bone pain. This variant typically develops in the metaphysis of long bones, particularly the distal femur and proximal tibia, extending from the metaphysis into the epiphysis in 83% of cases. However, it can also be found in less common sites, such as the ribs, mandible, and patella [107,108].

Telangiectatic osteosarcoma is biologically aggressive, characterized by a bone matrix filled with large blood-filled spaces. Consequently, its radiographic hallmark is extensive osteolysis with a multiloculated appearance [12,18] (Figure 19).

Telangiectatic osteosarcoma typically appears radiographically as a large, multilocular, expansile lesion with partially geographic bone loss and occasional sclerotic borders. It sometimes displays a well-defined margin with an irregular sclerotic border, alongside areas of a wide transitional zone and endosteal scalloping. A classic permeative pattern of bone destruction is rarely observed. The periosteal reaction may be present, although less vigorous than in the conventional variant, often manifesting as a Codman’s triangle, with the sunburst appearance being rare. The neoplastic matrix often contains small, faint radiopaque spots at the lesion’s periphery, and cortical bone interruption is frequent, allowing the lesion to extend into the surrounding soft tissues. The tumor frequently causes aneurysmal expansion of the cortical bone, though simple cortical expansion is less common. The radiographic appearance of telangiectatic osteosarcoma can resemble that of an aneurysmal bone cyst, challenging the diagnostic acumen of radiologists (Figure 20) [92]. Pathologic fractures occur in approximately 25% to 30% of cases, and soft tissue masses are also encountered [18]. It can be observed that there is no solid component within the aneurysmal bone cyst, whereas there is an enhanced solid component in telangiectatic osteosarcoma (Figure 19D).

CT imaging reveals a heterogeneous mass, often septate, characterized by low central attenuation due to necrotic tissue, along with cortical bone destruction [18]. Small peripheral calcific foci can also be detected. On T2-weighted imaging, telangiectatic osteosarcoma commonly shows heterogeneous signal intensity, while T1-weighted sequences often reveal areas of high signal due to methemoglobin. Fluid–fluid levels, though more typical of aneurysmal bone cysts, appear in fewer than half of telangiectatic osteosarcoma cases. MRI also highlights thick irregular septa at the lesion’s edges with a solid enhancing component that represents the vital tissue and may contain the bone matrix, showing strong post-contrast enhancement. Nodular intralesional thickening, aggressive growth, and osteoid matrix mineralization in a soft tissue mass aid in differentiating it from aneurysmal bone cysts [19,92].

#### 3.1.4. Small-Cell Osteosarcoma

Small-cell osteosarcoma accounts for approximately 1% of all osteosarcoma cases and typically affects individuals in their second and third decades of life, with a mild female predilection [95,96]. The 5-year survival rate for patients with small-cell osteosarcoma is approximately 42–50%, which is slightly lower than that for conventional osteosarcoma (53–61%) and Ewing sarcoma (51%) [109]. These tumors most commonly originate in the metaphyseal region of long bones, with the femur being the most frequently affected site. However, about 14% of cases are found in the diaphysis. Small-cell osteosarcoma can mimic Ewing sarcoma or primitive neuroectodermal tumors due to small round cells with hyperchromatic nuclei and minimal pleomorphism. Identifying osteoid matrix production is critical for diagnosis. Additionally, the absence of the EWS-ETS gene rearrangement helps differentiate it from Ewing sarcoma [89,110]. Therefore, identifying osteoid matrix production by the tumor cells is crucial for an accurate diagnosis. Additionally, it is important to rule out the EWS-ETS chromosome 22 rearrangement characteristic of Ewing sarcoma, as osteosarcoma and Ewing sarcoma can share similar histological features. Radiographically, small-cell osteosarcoma lacks distinctive features, presenting with a range of patterns from predominantly lytic to exuberant osteoid formation (Figure 21) [18,111]. This is often accompanied by a soft tissue mass and a vigorous periosteal reaction observed in more than 50% of cases [111].

#### 3.1.5. Low-Grade Central Osteosarcoma

Low-grade central osteosarcoma, also known as low-grade intramedullary osteosarcoma, is a rare, well-differentiated, and less aggressive variant of osteosarcoma that originates within the medullary cavity of bones, representing less than 2% of all osteosarcoma cases [18,94]. Among osteosarcoma subtypes, it typically presents with the most benign radiographic appearance overall [112]. Accurate diagnosis is crucial, as its treatment differs significantly from that of benign lesions. Therefore, radiologists should focus on identifying the subtle focal areas of aggressive features, which are the keys to diagnosis. Low-grade central osteosarcoma typically presents between the third and fourth decade of life and most often affects the meta-diaphyseal region of long bones, particularly the femur and tibia [112]. Clinically, patients usually present with swelling and mild symptoms, leading to delayed diagnosis when the tumors are already large. Compared to conventional osteosarcoma, patients with this variant generally have a more favorable prognosis, with a 5-year survival rate of 90% [113]. However, inadequate surgical treatment with insufficient margins can lead to the chance of local recurrence and metastases [114]. At histology, low-grade central osteosarcoma consists of a micro trabecular bone matrix within an unremarkable fibrous tissue with varying levels of bone formation. Low-grade central osteosarcoma shares histologic similarities with fibrous dysplasia and fibrous bony lesions but closely mirrors the microscopic features of low-grade parosteal osteosarcoma of which it is considered the intramedullary counterpart [112]. The key distinguishing factor between low-grade central osteosarcoma and benign fibrous lesions is the infiltration of tumor cells into mature bone trabeculae or cortical bone. The imaging features of low-grade central osteosarcoma are variable. In some case series, a prevalence of the lytic pattern was observed (60%), characterized by penetrating and slightly nuanced osteolysis with various degrees of trabeculation [112]. In others, the mixed pattern prevailed in up to 80% of the cases with a coexistence of lytic and sclerotic features [94]. The differential radiologic diagnoses include benign fibro-osseous lesions such as fibrous dysplasia, non-ossifying fibroma, and desmoplastic fibroma [94]. Cortical bone destruction or expansion is frequently observed, and while a periosteal reaction may occur, it is not consistently present [112]. The presence of aggressive imaging features such as cortical breach, soft tissue extension, and periosteal reaction is a helpful clue for the differentiation of low-grade central osteosarcoma from benign fibro-osseous lesions, as these features are unusual in benign lesions [112].

### 3.2. Juxtacortical (Surface) Osteosarcomas

#### 3.2.1. Parosteal Osteosarcoma

Parosteal osteosarcoma is a rare low-grade form of osteosarcoma that develops on the surface of the bone and is the most common type of juxtacortical osteosarcoma [18].

It typically affects individuals in their 30 s and 40 s with a slight predominance in females [97]. Parosteal osteosarcoma is usually characterized by slow growth, delayed metastasis, and favorable prognosis, with a 5-year survival rate of approximately 90% [115]. The tumor’s indolent nature often leads to late detection, but it responds well to surgical resection, contributing to the high survival rate [115]. Parosteal osteosarcoma arises in the outer fibrous layer of the periosteum [18]. Histologically, parosteal osteosarcomas have a dual appearance, being characterized by fibroblastic spindle cell stroma with minimal atypia and a large amount of mature trabecular bone commonly organized into long parallel strands [89,97]. At imaging, the trabecular bone matrix may appear similar to a hair-on-end or sunburst periosteal reaction outside the periosteum, although it is not strictly a periosteal reaction, as the tumor originates from the periosteum itself. In 90% of cases, parosteal osteosarcoma occurs in the metaphyseal region, with a strong preference for the distal femoral meta-diaphysis, particularly along the posterior cortex (62%) [116]. Plain radiography is usually the primary diagnostic tool, while CT is superior for evaluating cortical integrity [18,97]. MRI is more effective for assessing bone marrow invasion, soft tissue involvement, areas of the chondroid matrix, and potential sites of dedifferentiation [18,97]. In its early stages, parosteal osteosarcoma may resemble cortical thickening, leading to potential misdiagnosis as a healing stress fracture. About one-fourth of cases are initially seen with a cartilage cap on the outer surface of the tumor simulating an osteochondroma [18,97]. However, in parosteal osteosarcoma, the cartilage cap is typically irregular, thick, and incomplete, whereas in osteochondroma, it appears smooth, thin, and continuous [117]. Moreover, parosteal osteosarcoma lacks corticomedullary continuity between the tumor and the underlying medullary canal [18]. As the lesion grows, it often presents as a lobulated, cauliflower-like exophytic mass protruding from the underlying cortex, with a broad base of attachment (Figure 22). The bone formation progresses from the center toward the periphery, with an irregular mineralization pattern; therefore, the periphery of the tumor is generally less dense compared to the center. At the time of diagnosis, the tumor can range in size from 2 cm to over 10 cm, often presenting with a fusiform shape that extends along the length of the bone and sometimes wraps circumferentially around it.

The underlying bone cortex may show thickening due to focal expansion of the inner part of the tumor and its fusion with the cortex. In more advanced stages, the cortex may be partially eroded. The tumor commonly manifests as a partially circumferential thickening of the cortex, exhibiting a disorganized mix of osteolytic areas (bone loss) and regions of bone thickening. The most characteristic feature of the tumor is a lobulated exophytic lesion with dense ossification at its core.

CT imaging can reveal a cleavage plane, often referred to as the “string sign”, between the tumor and the normal cortex in approximately 65% of cases. This feature corresponds histologically to the inner layer of the periosteum, which is interposed between the tumor mass and the cortical bone [116,118,119].

MRI is the primary imaging modality for assessing intramedullary involvement and the tumor’s extension into peri-skeletal soft tissues in parosteal osteosarcoma [97]. The ossified component of parosteal osteosarcomas typically shows a low signal on both T1- and T2-weighted images, resembling the appearance of cortical bone. Post-contrast images generally show intense heterogeneous enhancement. Medullary extension occurs in 22% to 58% of patients, though this finding alone does not have significant prognostic implications [116,120,121]. The presence of an ill-defined enhancing soft tissue mass within or adjacent to the ossified tumor is suggestive of areas of dedifferentiation. The enhancing region may serve as the target for biopsy to assess the high-grade component [122]. Dedifferentiation into high-grade osteosarcoma has been reported in up to 43% of cases [120,123,124]. The process is histologically characterized by the coexistence of low-grade parosteal osteosarcoma and higher-grade osteosarcoma. Dedifferentiation is associated with increased bone lysis and the presence of soft tissue components without ossification [97]. Areas of unmineralized soft tissue greater than 1 cm^3^ and regions with fluid–fluid levels are indicative of dedifferentiation to a high-grade lesion [18,116]. The extension of the tumor into surrounding soft tissues is often inferred by the displacement of fatty tissue planes on MRI. Differential diagnoses include benign lesions such as osteochondroma, periosteal chondroma, and malignant lesions such as periosteal chondrosarcoma, fibrous malignancies, and other subtypes of superficial osteosarcomas [12,18,20].

#### 3.2.2. Periosteal Osteosarcoma

Also referred to as juxtacortical chondroblastic osteosarcoma, this rare form of intermediate-grade surface osteosarcoma represents less than 2% of all osteosarcoma cases and is the second most common surface-based osteosarcoma, following parosteal osteosarcoma [18,98]. Histologically, periosteal osteosarcomas are characterized by regions of cartilage transitioning into an osteoid matrix, often displaying a combination of cartilaginous and osteoid tissue [97,98]. Periosteal osteosarcoma shows a slight male predominance, with the condition typically occurring in individuals in their second or third decade of life. It often presents with minimal to no pain initially, followed by the development of symptoms such as swelling and/or pain that persist for weeks to months [18]. The prognosis (83% 5-year survival rate) is better than that for conventional but worse than that for parosteal osteosarcoma [125]. Periosteal osteosarcoma is most found in the diaphysis or meta diaphysis of long bones, particularly affecting the tibia and femur (85–95%), while the ulna and humerus are less frequently involved (5–10%). It is rarely found in the jaw, clavicle, pelvis, rib, and skull [118]. Unlike the parosteal variant, which originates from the outer layer of the periosteum, periosteal osteosarcoma typically arises from the inner cambium layer of the periosteum and is attached to the underlying cortex. At diagnosis, these tumors usually measure less than 5 cm and have a fusiform shape, developing along the cortical surface.

#### 3.2.3. High-Grade Surface Osteosarcoma

High-grade surface osteosarcoma is a rare subtype, accounting for only 0.4% of all osteosarcomas and representing the least frequent form of juxtacortical osteosarcoma [95]. It usually occurs in the second to third decades of life, primarily affecting the diaphyses and metaphyses of long bones, with the femur being the most common site [126]. The prognosis has historically been considered like that of conventional osteosarcoma, with a 5-year survival rate of 46.1%, but recent studies indicate improved outcomes due to aggressive chemotherapy and surgical resection [126].

Radiographically, high-grade surface osteosarcoma is characterized by dense ossification and periosteal reactions in most cases [124]. Cortical thickening and erosion are frequently observed, and in some cases, the tumor invades the medullary cavity, with reports of medullary involvement varying from 8% to 48% [126]. Radiographs often show cortical irregularity with areas of sclerosis, periosteal lifting, and various patterns of ossification, ranging from dense ossification to mixed lytic and sclerotic changes [18]. Tumor sizes can vary widely at the time of the diagnosis, ranging from 4.5 to 22 cm [120,124], with dense ossification and cortical erosion being common findings [18,120].

CT imaging provides further detail, often revealing cortical destruction, thickening, and periosteal reaction. These periosteal reactions may present as Codman’s triangle, a sunburst appearance, or spiculated periosteal responses. Invasion into the medullary cavity can be visualized as areas of medullary sclerosis or focal bone destruction within the marrow space [18]. The extensive circumferential bone invasion could help differentiate high-grade form other types of superficial osteosarcomas.

## 4. Recent Advancements

Radiomics and artificial intelligence (AI) are emerging as powerful tools in the study and management of osteosarcoma and bone tumors in general, offering new approaches to both diagnostic and predictive tasks [19,127]. Radiomics involves the extraction of high-dimensional quantitative features from imaging data imperceptible to the human eye. These features are then analyzed using advanced machine learning and deep learning algorithms to derive information and develop diagnostic and predictive models [128,129,130].

Currently, AI algorithms in bone tumors are applied across a wide range of tasks, including detection, segmentation, and volume calculation, classification, grading, the assessment of the tumor necrosis rate in both primary and metastatic bone tumors, and predictive models for therapy response, as well as prognostic tasks related to survival recurrence and metastasis [131]. Depending on the specific task, these models rely on various radiological modalities (such as X-ray, CT, MRI, and SPECT [27,131,132]. However, recent applications based on X-ray imaging are particularly important for early detection, as X-ray is typically the first-line imaging modality [132]. An important application of AI is the ability to integrate clinical data with imaging [131,132]. Modern approaches that combine clinical metadata such as age, affected bone site, tumor position, and gender with X-ray imaging using multimodal deep learning models can significantly enhance the classification accuracy of primary bone tumors [132].

In osteosarcoma, radiomics has been extensively applied to MRI and CT imaging, focusing on the pre-treatment prediction of histologic response and long-term outcomes, such as overall survival and event-free survival [133,134,135]. Studies using T2-weighted imaging (T2-WI), apparent diffusion coefficient (ADC) maps, and contrast-enhanced T1-weighted imaging have demonstrated encouraging results for survival prediction [19]. The combination of pre- and post-chemotherapy radiomics data enhances the accuracy of baseline models in predicting treatment response [19]. A recent study by White et al. demonstrates that T2-weighted MRI radiomic models show promising potential as prognostic biomarkers for predicting tumor response to neoadjuvant chemotherapy and clinical outcomes, including overall survival and disease-free survival [134]. Another recent study by Zhang et al. demonstrated on a large cohort of 209 patients that a nomogram combining MRI diffusion-weighted imaging (DWI) radiomics and clinical features outperforms standalone models, offering enhanced accuracy in predicting neoadjuvant chemotherapy response in osteosarcoma and showing significant potential for clinical application [136].

AI algorithms, particularly convolutional neural networks (CNNs), have further streamlined radiomics workflows by automating tasks like tumor segmentation, which is critical for feature extraction [130,137,138]. Modern versions of convolutional neural networks, such as the multiple supervised fully convolutional networks, have demonstrated superior performance in tumor segmentation tasks, providing a fast and accurate delineation of osteosarcoma boundaries, which can aid in more precise treatment planning [139]. These automated approaches not only save time but also reduce inter-observer variability, especially for tumors with irregular shapes or challenging anatomical locations.

Despite these advancements, significant challenges persist, as evidenced by a recent systematic review highlighting the limitations of many radiomics studies on osteosarcoma, which consequently leads to limited clinical applications [27,140]. The systematic review by Zhou et al. in 2021 included many of the available studies on radiomic applications in osteosarcoma, highlighting the methodological issues in this research area [27]. Many radiomics studies are based on retrospective heterogeneous datasets with limited external validation. The limited sample sizes in osteosarcoma research pose a significant challenge in developing robust AI models and the disease’s rarity restricts access to high-quality annotated data. Additionally, the lack of standardized radiomics pipelines and reporting practices complicates the comparison and reproducibility of findings across studies. Guidelines like CLAIM (Checklist for Artificial Intelligence in Medical Imaging) and, more recently, METRICS (METhodological Radiomics Score), are increasingly recommended to address these gaps and ensure rigorous study design and reporting [27,141].

Looking ahead, collaborative efforts to create multi-institutional databases could help overcome the limitations of small sample sizes and improve the generalizability of AI models [140]. In addition, high-quality expert annotations have been shown to enhance the accuracy of artificial intelligence models for osteosarcoma X-ray diagnosis [142].

Moreover, the integration of radiomics with clinical, genomic, and histopathological data could unlock new dimensions of personalized medicine, tailoring treatment strategies to the unique biological and imaging profiles of individual patients [143]. As AI and radiomics continue to evolve, they hold great promise in transforming the landscape of osteosarcoma care, enabling more precise data-driven decisions for both clinicians and patients.

## 5. Conclusions

Osteosarcomas are rare but aggressive primary bone tumors requiring accurate diagnosis and comprehensive imaging for effective management. Although uncommon, they are the most frequent malignant bone tumors in children and adolescents, highlighting the need for early detection. The Bone-RADS classification aids radiologists in the initial risk stratification of bone lesions, often detected incidentally during radiographic studies. Advances in imaging, particularly MRI and CT, are pivotal for diagnosing, staging, and monitoring overall survival, offering detailed insights into tumor characteristics and guiding treatment. Emerging technologies like artificial intelligence in radiomics and radiogenomics show promise in enhancing diagnostic capabilities, though their clinical application remains limited by data and methodological challenges. Despite technological progress, radiologists’ expertise remains essential for interpreting complex imaging and ensuring optimal patient outcomes.

## Figures and Tables

**Figure 1 cancers-17-00599-f001:**
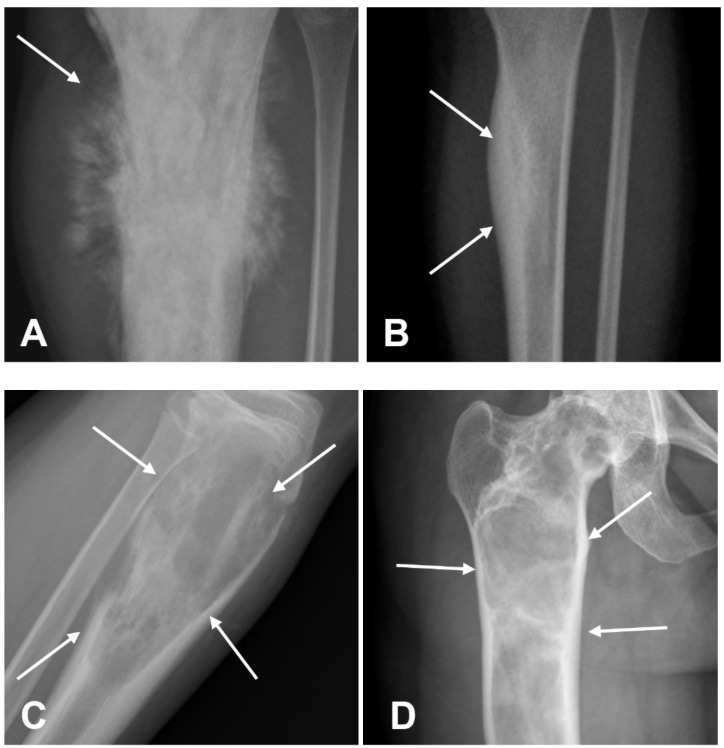
Radiographic examples of sclerotic ((**A**,**B**); arrows) and lucent ((**C**,**D**); arrows) lesions. Both sclerotic and lucent patterns can appear in both benign and malignant bone lesions. Note, for comparison, two examples of osteosarcomas (**A**,**C**) and benign lesion, osteoid osteoma (**B**), and fibrous dysplasia (**D**).

**Figure 2 cancers-17-00599-f002:**
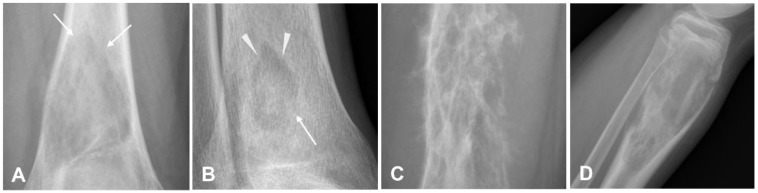
Examples of osteosarcomas with varying margin patterns. (**A**) Ill-defined margins (arrow) (Type II). (**B**) Mixed well-defined (arrowhead) and ill-defined (arrow) margins, also referred to as “changing margins” (Type IIIA). (**C**) Moth-eaten osteolytic lesion or (**D**) permeative osteolytic lesion (Type IIIB). A permeative bone process or moth-eaten appearance refers to multiple small endosteal lucent lesions or holes.

**Figure 3 cancers-17-00599-f003:**
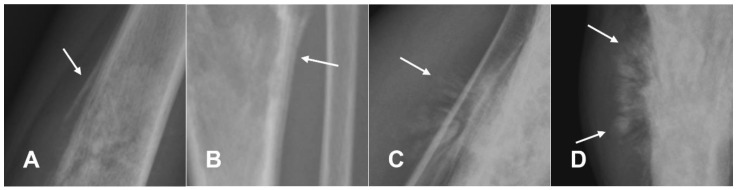
Various types of periosteal reactions. (**A**) Codman’s triangle (arrow); (**B**) lamellated or “onion skin” pattern (arrow); (**C**) “hair-on-end” appearance (arrow); (**D**) sunburst appearance (arrows).

**Figure 4 cancers-17-00599-f004:**
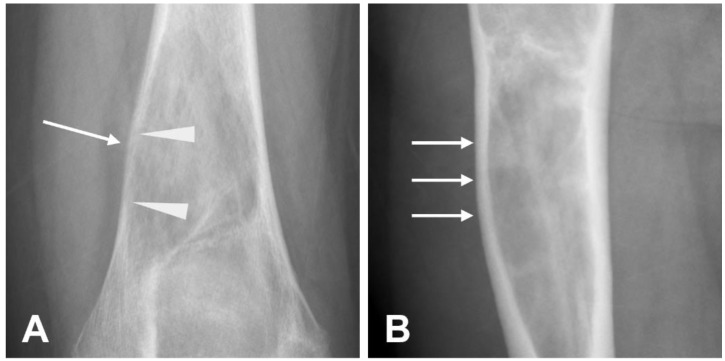
Examples of endosteal scalloping. (**A**) Thinning of the inner bony cortex (arrowhead) in a case of osteosarcoma, associated with mild cortical bulging (arrow). For comparison, (**B**) shows uniform cortical thinning (arrows) associated with a radiolucent lesion in case of fibrous dysplasia.

**Figure 5 cancers-17-00599-f005:**
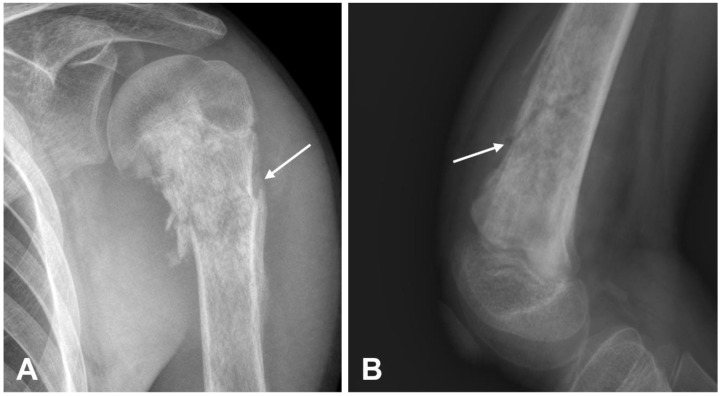
Example of pathological fractures in osteosarcomas. (**A**) The radiograph presents a meta-epiphyseal lesion of the proximal humerus (fibroblastic variant or conventional osteosarcoma) with mixed density, osteoid matrix mineralization, and permeative margins. A pathological fracture is also visible (arrow). (**B**) Another radiographic example shows a pathological fracture (arrow) associated with a sclerotic bone lesion in femur metaphysis.

**Figure 6 cancers-17-00599-f006:**
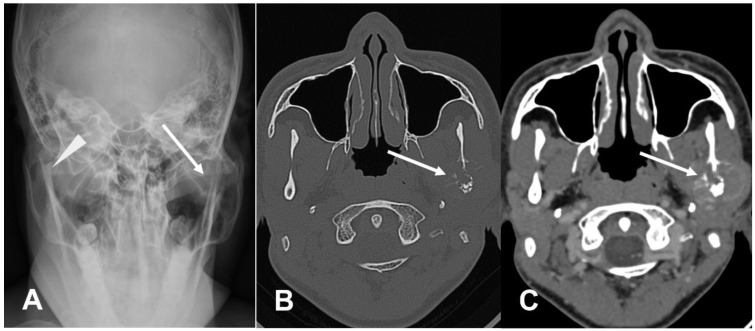
CT assessment of osteosarcoma in a complex anatomical region. On radiograph (**A**), the lesion is difficult to identify, but an asymmetry of the left mandibular ramus (solid arrow) compared to the normal contralateral side (arrowhead) can be observed in the frontal view. On CT (**B**,**C**), an osteolytic lesion with an internal osteoid matrix is clearly identified (arrows).

**Figure 7 cancers-17-00599-f007:**
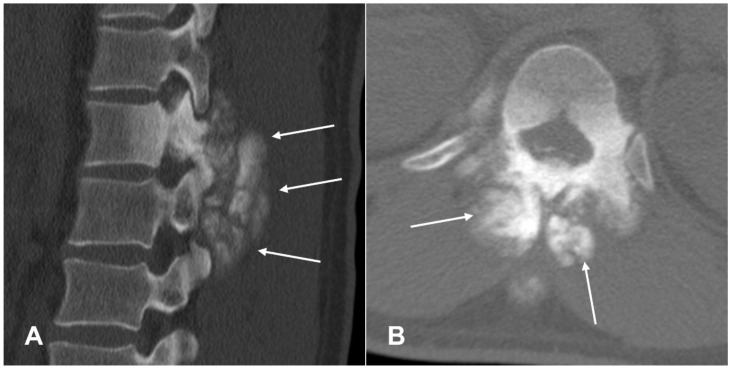
Vertebral osteosarcoma. Sagittal (**A**) and axial (**B**) CT scans of the spine, theosteoid matrix (arrows) can be seen within the soft tissue component of the tumor, which involves the adjacent paravertebral muscles.

**Figure 8 cancers-17-00599-f008:**
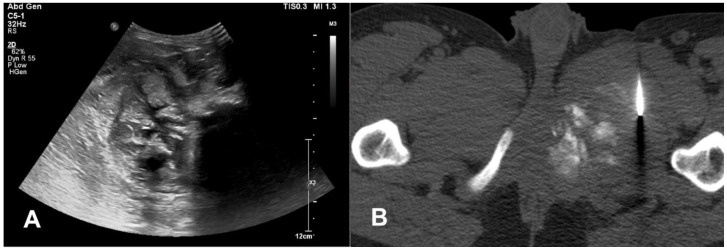
Biopsy of a suspected malignant bone lesion. (**A**) Preliminary assessment with ultrasound; (**B**) CT-guided biopsy.

**Figure 9 cancers-17-00599-f009:**
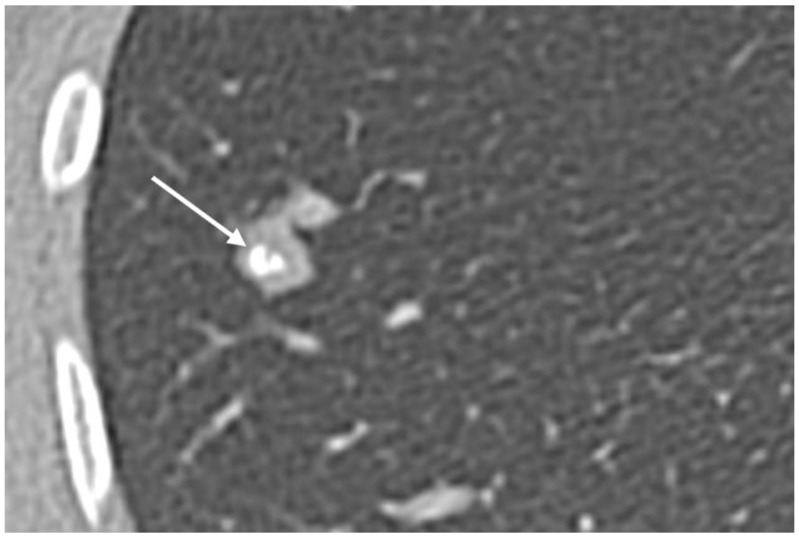
Pulmonary metastases from osteosarcoma with calcifications (arrow).

**Figure 10 cancers-17-00599-f010:**
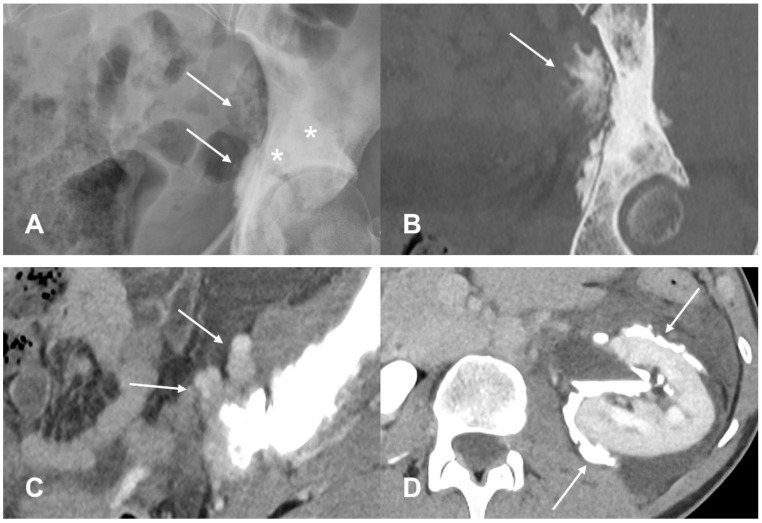
(**A**) Pelvic radiograph showing osteoblastic lesion (asterisk) at the left acetabulum with soft tissue component (arrow) and osteoid matrix, more clearly seen on CT with bone window in (**B**). (**C**) Soft tissue window on CT reveals the soft component (arrows). The ureter was observed to be infiltrated and indistinguishable within the tumor mass. (**D**) Hydronephrosis has caused rupture of renal pelvis with evident extravasation of contrast media (arrows).

**Figure 11 cancers-17-00599-f011:**
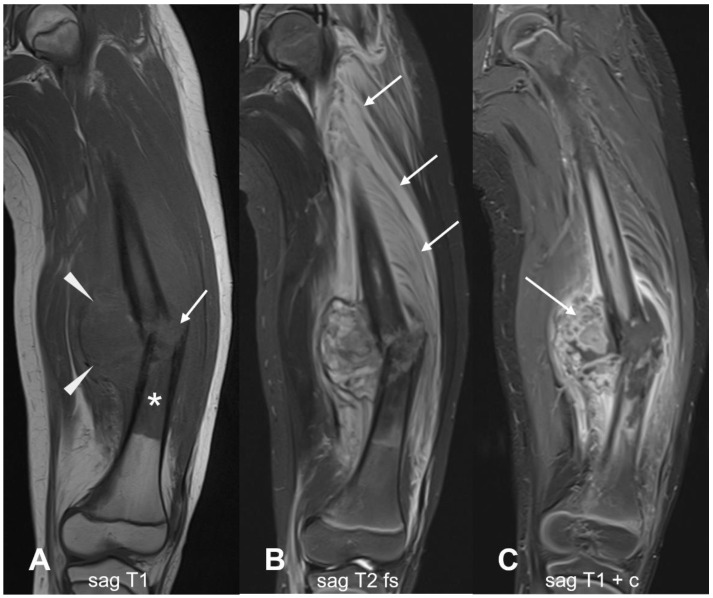
Typical example of osteosarcoma on MRI. (**A**) T1-weighted image. A pathological fracture (arrow) is noted along with an ill-defined hypointense mass (arrowheads) and intramedullary lesion at diaphysis (asterisk). (**B**) The T2 fs sequence demonstrates the lesion and the extent of edema, which in this case extends to the entire thigh musculature (arrows). (**C**) Post-contrast T1-weighted sequence, highlighting the proliferating pathological tissue (arrow) within the lesion. sag = sagittal; ax = axial; cor = coronal; T1 = T1-weighted; T2 = T2-weighted; fs = fat saturated; c = contrast-enhanced.

**Figure 12 cancers-17-00599-f012:**
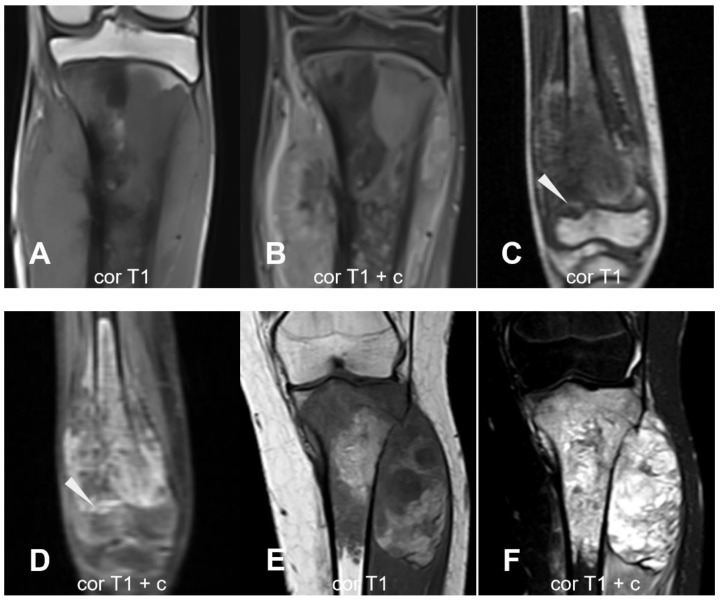
Epiphyseal involvement. (**A**,**B**) Lesion of the proximal tibial metaphysis. The intramedullary lesion abuts the epiphyseal plate superiorly, with the epiphysis preserved. (**C**,**D**) The lesion abuts the epiphysis with epiphyseal involvement (arrowhead). (**E**,**F**) Epiphyseal involvement. sag = sagittal; ax = axial; cor = coronal; T1 = T1-weighted; T2 = T2-weighted; fs = fat saturated; c = contrast-enhanced.

**Figure 13 cancers-17-00599-f013:**
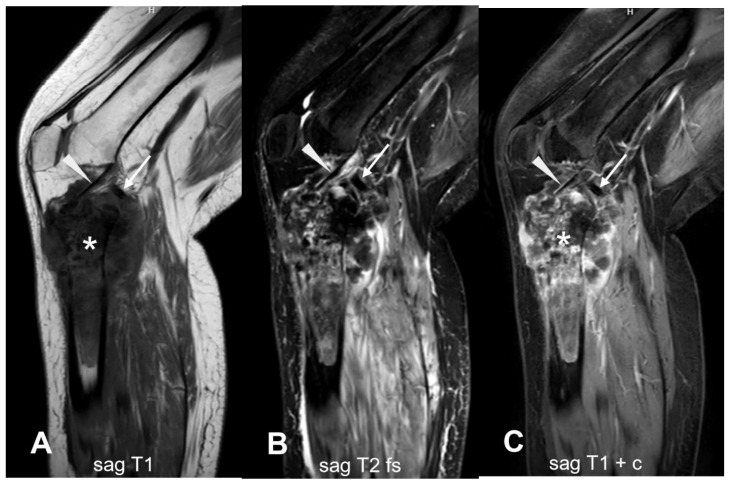
Joint involvement. (**A**–**C**) This case demonstrates an extensive osseous lesion (asterisk) in the right proximal tibia, extending from the epiphysis to the proximal one-third of the diaphysis, with cortical breach and a large soft tissue component. The mass appears heterogeneous with a hypointense signal on T1-weighted images and a variable signal on T2-weighted images, along with heterogeneous contrast enhancement. Intra-articular extension into the right knee joint is observed, encasing the anterior (arrowhead) and posterior cruciate ligaments (arrow). sag = sagittal; T1 = T1-weighted; T2 = T2-weighted; fs = fat saturated; c = contrast-enhanced.

**Figure 14 cancers-17-00599-f014:**
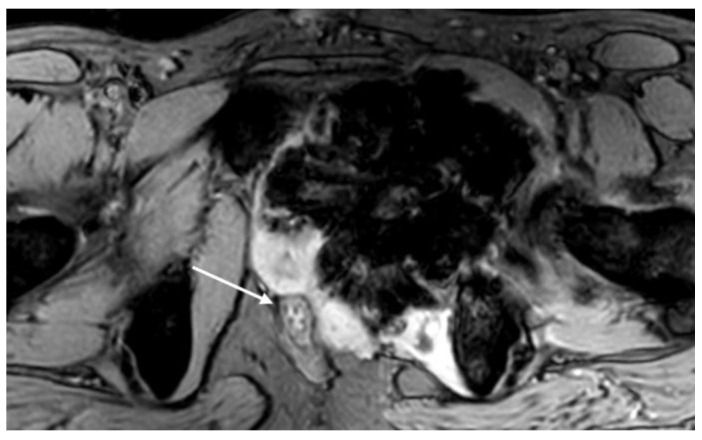
Soft tissue invasion. Critical concerns include the loss of fat planes with nearby organs (urinary bladder, prostate, rectum (arrow)), indicating possible tumor infiltration and the involvement of the adductor brevis, adductor magnus, and obturator internus muscles.

**Figure 15 cancers-17-00599-f015:**
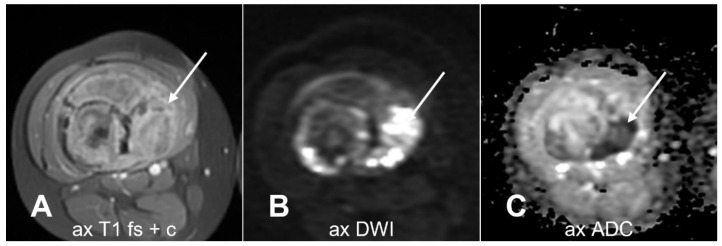
The area of highest cellularity (arrows) on post-contrast T1-weighted sequence (**A**) shows high signal intensity on DWI (**B**) and low signal intensity on ADC map (**C**), which serve as potential biopsy targets. ax = axial; T1 = T1-weighted; T2 = T2-weighted; fs = fat saturated; c = contrast-enhanced. Courtesy of Dr. Hammar Haouimi, licensed under CC BY-NC-SA 3.0.

**Figure 16 cancers-17-00599-f016:**
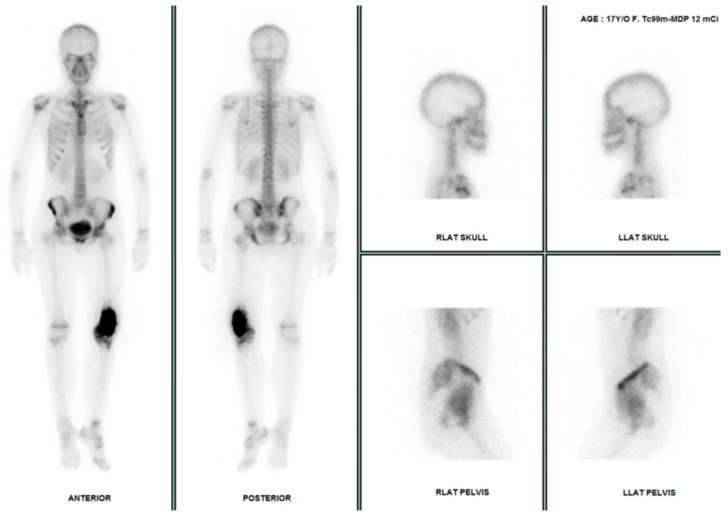
Bone scan. A 17-year-old woman with a suspected osteosarcoma at the left distal femur underwent a Tc-99 m MDP bone scan. The results show significantly increased uptake at the left distal femur, while the rest of the skeleton, epiphyseal plates, kidneys, and bladder appear unremarkable.

**Figure 17 cancers-17-00599-f017:**
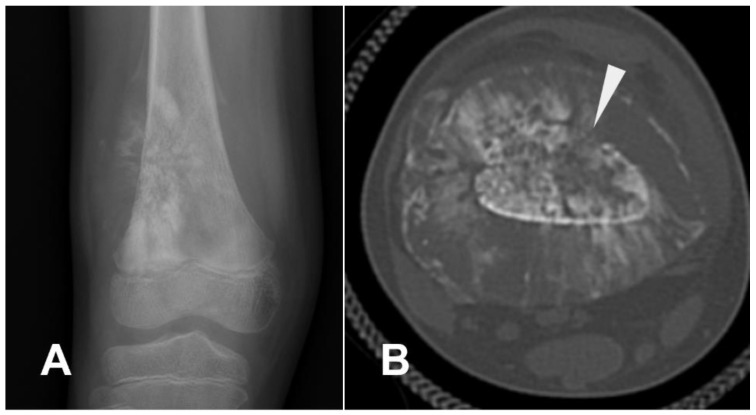
Two examples of osteoblastic osteosarcoma located in the right distal femur (**A**,**B**) and proximal tibia (**C**,**D**). Radiographs (**A**,**C**) show a sunburst periosteal reaction (arrowheads). CT images (**B**,**D**) reveal cortical destruction ((**B**), arrowhead) and cortical thickening ((**D**), arrowheads).

**Figure 18 cancers-17-00599-f018:**
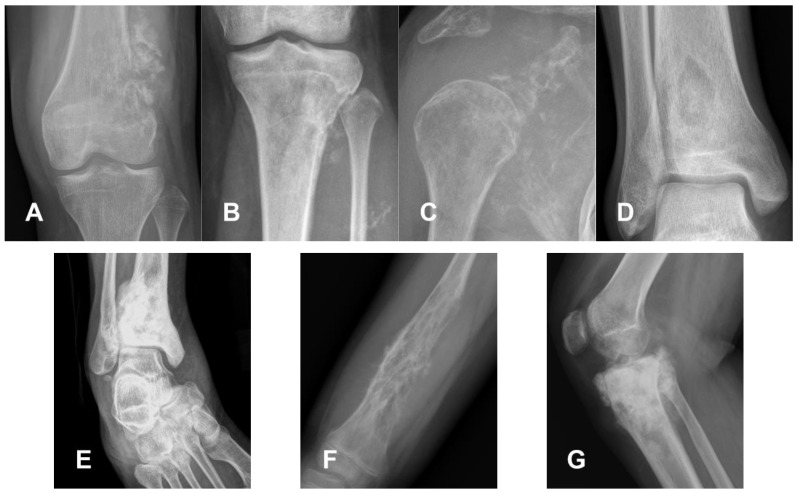
Radiographic appearances of conventional osteosarcoma variants. (**A**,**B**) Chondroblastic variant; (**C**,**D**) fibroblastic variant; (**E**) fibroblastic variant; (**F**) mixed chondroblastic and fibroblastic variant; (**G**) mixed osteoblastic and chondroblastic variant. Significant overlap exists among subtypes, with typically mixed density of the lesion, as in (**A**). Predominantly lytic forms are noted in (**C**,**F**), while sclerotic forms are evident in (**E**,**G**). When mineralization is present, it is primarily osteoid, as shown in (**E**,**G**).

**Figure 19 cancers-17-00599-f019:**
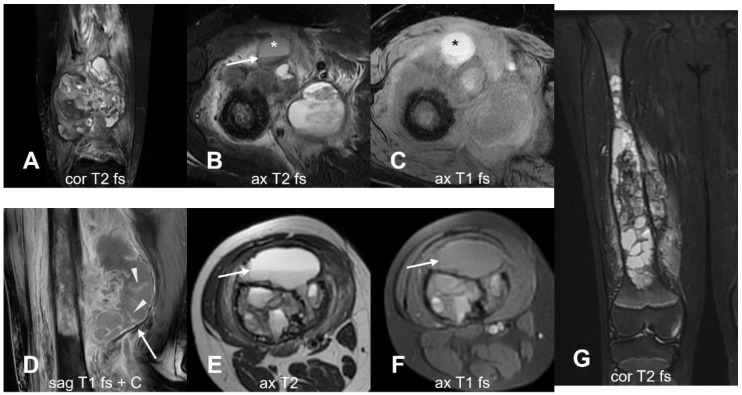
Telangiectatic osteosarcoma of the distal femoral diaphysis. (A) The lesion demonstrates bone destruction and a multiloculated soft tissue component. Regions of fluid–fluid levels are evident ((**B**), arrow) along with areas of high signal intensity on T2-weighted image ((**B**), asterisk) and T1-weighted images with fat suppression ((**C**), asterisk), indicative of hemorrhage. (**D**) Post-gadolinium images show peripheral (arrow) and septal (arrowheads) enhancement. (**E**,**F**) Fluid–fluid levels are visible (arrow), showing different stages of hemoglobin degradation compared to the previous case (arrow). (**G**) Another example of telangiectatic osteosarcoma with extensive involvement of the femoral diaphyseal marrow. Image courtesy of Dr. Yasser Asiri, licensed under CC BY-NC-SA 3.0.

**Figure 20 cancers-17-00599-f020:**
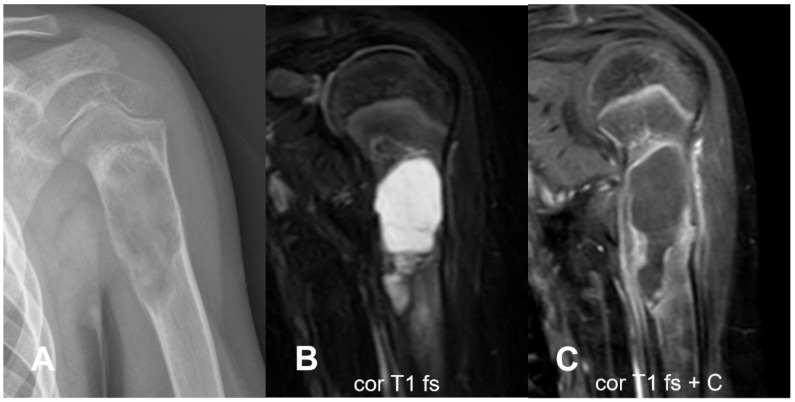
Aneurysmal bone cyst. (**A**) Radiograph shows a lytic lesion in the proximal humeral metaphysis with a narrow zone of transition, causing mild cortical bulging. (**B**) T2-weighted fat-saturated sequences reveal fluid signal within the cyst. (**C**) Mild perilesional enhancement is observed in post-contrast sequences.

**Figure 21 cancers-17-00599-f021:**
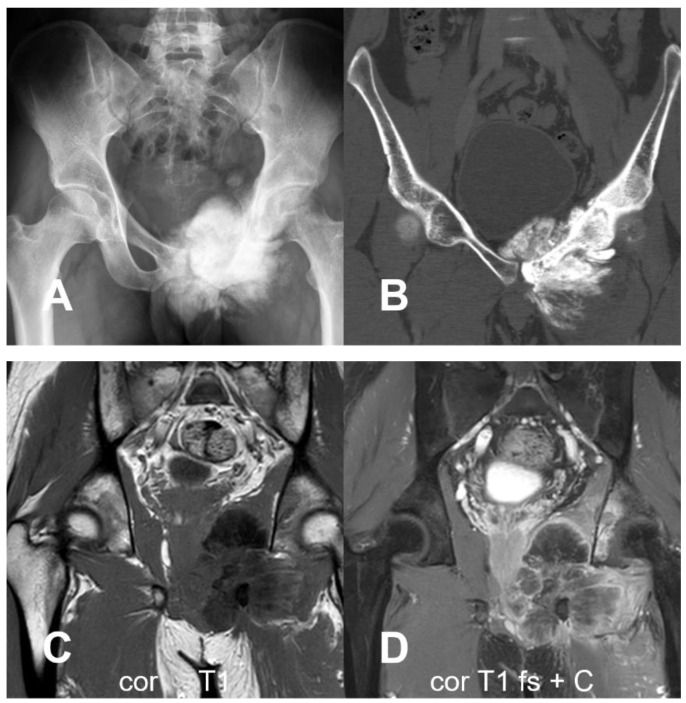
Small-cell osteosarcoma of the iliopubic branch, presenting as a sclerotic lesion with exuberant osteoid matrix formation. The osteoid matrix is clearly visible, appearing as an abnormal area of increased bone density area on radiographs (**A**) and CT (**B**) and as a hypointense alteration on T1-weighted MRI (**C**,**D**).

**Figure 22 cancers-17-00599-f022:**
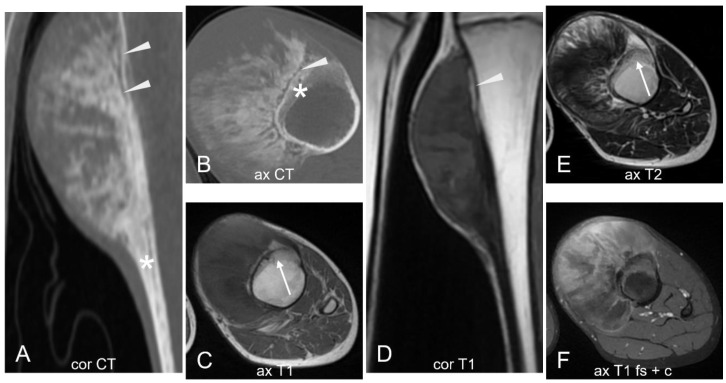
Parosteal osteosarcoma. A large exophytic mass protrudes from the anteromedial cortex of the proximal tibial metaphysis. On CT (**A**,**B**), cortical thickening (asterisks) and a dense osteoid matrix are visible, with greater density at the center compared to the periphery. A thin cleavage plane between the cortex and the tumor ((**A**,**B**,**D**), arrowheads) is seen, while adjacent cortical thickening and sclerosis are noted in the older areas, along with some zones of cortical lysis ((**C**,**E**), white arrows). The lesion has well-defined margins and appears hypointense on T1 (**C**,**D**), predominantly hypointense on T2 (**E**) with heterogeneous post-contrast enhancement (**F**). No cystic areas or fluid–fluid levels are identified. Courtesy of Dr. Dalia Ibrahim, Lecturer, Cairo University. Licensed under CC BY-NC-SA 3.0.

**Table 1 cancers-17-00599-t001:** Role of different imaging modalities in the assessment of osteosarcoma.

Radiograph	CT	MRI	Bone Scan and PET/CT
- First step in the evaluation of bone lesions, essential to guide diagnosis and follow-up (Bone-Rads)	- Detailed anatomical visualization of challenging areas (maxillo-facial bones, skull, vertebral spine) - Better visualization of tumor mineralization, small soft tissue calcifications, cortical changes, and periosteal reaction - Evaluation for pathologic fracture - Overall tumor staging (distant metastasis)	- Local staging of the tumor (tumor extension, epiphyseal and joint invasion, soft tissue involvement, neurovascular involvement, skip lesions) - Diagnostic insights based on the study of the tumor’s structure (fluid, restricted diffusion, vascularization, enhancement, etc.)	- Screening tool for primary lesions and distant bone metastasis

**Table 2 cancers-17-00599-t002:** Radiographic features evaluated for assigning the Bone-RADS score [32].

Feature Evaluated		Corresponding Points
Margins	IA (well defined, sclerotic)	1
	IB (well defined, non-sclerotic)	3
	II (ill defined)	5
	IIIA (changing margins from well defined to ill-defined) IIIB (moth-eaten or permeative) IIIC (radiographically occult with invisible margins)	7
Periosteal reaction	none	0
	non-aggressive	2
	aggressive	4
Endosteal reaction	mild	0
	moderate	1
	deep	2
Pathological fracture	no	0
	yes	2
Soft tissue mass	no	0
	yes	4
Known primary cancer	no	0
	yes	2
Total Points		0–21

**Table 3 cancers-17-00599-t003:** Risk stratification according to Bone-RADS; increasing points correspond to an increased Bone-RADS score and risk of malignancy.

Total Points	Bone RADS Score	Description	Lesions Meaning and Recommendations
N/A	0	incompletely characterized	additional imaging is needed
1–2	1	very low risk	very likely benign pathognomonic benign lesion
3–4	2	low risk	probably benign if asymptomatic: radiographic follow-up if symptomatic: treatment
5–6	3	intermediate risk	potentially malignant orthopedic oncology referral for biopsy and treatment planning
≥7	4	high risk	malignant until proven otherwise orthopedic oncology referral for biopsy and treatment planning

N/A = not applicable.

**Table 4 cancers-17-00599-t004:** Osteosarcoma subtypes.

Subtype		Prevalence	Histology
Intramedullary (Central)	Conventional (high-grade) osteosarcoma	80% [1,4,12]	Spindle-to-polyhedral-shaped malignant mesenchymal cells with high cellularity, nuclear polymorphism, atypia Extracellular matrix production may be osteoblastic, osteoclastic, fibroblastic, or a combination
	Telangiectatic osteosarcoma	2–12% [92,93]	Dilated hemorrhagic sinusoids and small amounts of osteoid
	Low-grade osteosarcoma	<2% [18,94]	Well-differentiated cells embedded in the osseous matrix and fibrous stroma, with small amounts of osteoid
	Small-cell osteosarcoma	1.5% [95,96]	Numerous small round malignant cells within an osteoid matrix Due to small round cells with hyperchromatic nuclei, it can be confused with Ewing sarcoma or primitive neuroectodermal tumor
Juxtacortical (Surface)	Parosteal osteosarcoma	1–5% [18,97]	Low-grade with a well-differentiated mostly cartilaginous matrix with minimal osteoid
	Periosteal osteosarcoma	1–2% [18,98]	Mostly cartilaginous matrix; minimal osteoid
	High-grade osteosarcoma	<1% [18,95]	High-grade spindle-shaped cells with nuclear pleomorphism

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
