# Peer review of "Multimodal Imaging of Osteosarcoma: From First Diagnosis to Radiomics"

_cancers, 2025, doi:10.3390/cancers17040599_

Round 1

Reviewer 1 Report

Comments and Suggestions for Authors

The paper provides a comprehensive and in-depth review of osteosarcoma imaging, addressing various modalities such as conventional radiography, CT, MRI, and emerging technologies like radiomics and artificial intelligence (AI). It is academic and technical in nature, targeting radiologists, oncologists, and researchers in the field of imaging sciences. Its primary focus is to enhance the understanding and clinical practice in diagnosing osteosarcoma through the use of multimodal imaging.

The detailed discussion of diagnostic challenges, risk stratification (Bone-RADS), and subtype-specific imaging characteristics significantly enhances the paper’s clinical value. Moreover, the extensive use of figures, tables, and radiological examples effectively clarifies complex imaging concepts, such as periosteal reactions and differential diagnoses. These visual aids strengthen the paper’s role as an educational resource for specialists.

Minor Suggestions for Improvement:

1. Accessibility for Non-Specialists:

The highly specialized language and technical depth might limit the paper’s accessibility to general practitioners or professionals outside the field of radiology. Consider simplifying certain sections or including a glossary of technical terms to broaden its audience.

2. Balanced Emphasis:

While the focus on radiomics and AI is forward-looking and valuable, it may overshadow other well-established diagnostic approaches. Expanding the discussion of traditional methods in relation to these innovations could provide a more balanced perspective.

Author Response

Comment 1: The highly specialized language and technical depth might limit the paper’s accessibility to general practitioners or professionals outside the field of radiology. Consider simplifying certain sections or including a glossary of technical terms to broaden its audience.

Response 1: Thank you for pointing this out. We use the medical terms, which are usually used in daily clinical practice.  So, we do not provide a glossary of technical terms in this research.

Comment 2: While the focus on radiomics and AI is forward-looking and valuable, it may overshadow other well-established diagnostic approaches. Expanding the discussion of traditional methods in relation to these innovations could provide a more balanced perspective.

Response 2: Agree. We have revised it to emphasize this point. Mention in section number 4, pages 27-28.

Reviewer 2 Report

Comments and Suggestions for Authors

This paper provides a comprehensive review of the different types of osteosarcoma and in-depth discussions of some of the prominent imaging techniques employed to diagnose and assess cancer disease. The paper can be further improved before publication.

1)      Introduction: I suggest for authors to outline the key sections of the paper towards the end of the introduction section. This helps readers to identify the key parameters/components/domains reviewed at the start of the paper.

2)      Figure 6: The arrows are missing in B and C.

3)      Section 3: It should be Table 4 instead of Table 3.

4)      New Perspectives: When I read the abstract, it gave me the impression that some machine learning studies would be reviewed. However, this section merely touches the general insights of AI-related domains. I think the strength of the paper would increase if the authors include some discussions of the machine learning studies found in literatures, highlighting the issues that present challenges and maybe offer some insights on the way forward.

Author Response

Comment 1: Introduction: I suggest for authors to outline the key sections of the paper towards the end of the introduction section. This helps readers to identify the key parameters/components/domains reviewed at the start of the paper.

Response 1: Thank you for pointing this out. We agree with this comment. Therefore, we have revised our Simple Summary and Abstract on pages 1-2.

Comment 2: Figure 6: The arrows are missing in B and C.

Response 2: Thank you for pointing this out. We already fixed it, on page 8.

Comment 3: Section 3: It should be Table 4 instead of Table 3.

Response 3: Thank you for pointing this out. We already fixed it, on page 17

Comment 4: New Perspectives: When I read the abstract, it gave me the impression that some machine learning studies would be reviewed. However, this section merely touches the general insights of AI-related domains. I think the strength of the paper would increase if the authors include some discussions of the machine learning studies found in literatures, highlighting the issues that present challenges and maybe offer some insights on the way forward.

Response 4: Thank you for pointing this out. We agree with this comment. Therefore, we have revised our manuscript in section number 4, pages 27-25.

Reviewer 3 Report

Comments and Suggestions for Authors

Concise information on multimodal approach How are CT, MRI, Bone Scan and PET/CT integrated in the multimodal approach and how does each modality uniquely contribute to the diagnosis and staging of osteosarcoma?

While the Bone-RADS classification system is based on qualitative findings, the supplementary information enhances the contextual understanding of lesions seen in practice. Expand on how it helps to stratify risk and guide the management of osteosarcoma based on imaging findings.

Exploration of utilization of artificial intelligence and radiomics in osteosarcoma diagnosis Explain how these technologies improve the analysis and interpretation of imaging data in detail.

Explain the difficulties in early-stage diagnosis of osteosarcoma and how imaging strategies are adapted to maximize early detection.

Explain how MRI is used to determine the extent of a tumor and how the different MRI sequences can provide information about different tumor features.

Discuss the implications of the presence of pathological fractures on the prognosis of osteosarcoma and their influence on treatment plans.

They use novel imaging modalities, such as dynamic contrast-enhanced MRI and diffusion-weighted imaging, to evaluate tumor biology and treatment response.

Discuss the implications of the challenges of current imaging techniques for identifying skip lesions and how these factors impact surgical planning and prognosis.

More relevant papers, especially those on advanced imaging technologies and artificial intelligence methods in bone tumor diagnosis, need to be introduced to improve the introduction and literature review.

1. Huang, H., Huang, F., Liang, X., Fu, Y., Cheng, Z., Huang, Y.,... Chen, Y. (2023). Afatinib Reverses EMT via Inhibiting CD44-Stat3 Axis to Promote Radiosensitivity in Nasopharyngeal Carcinoma. Pharmaceuticals, 16(1), 37. doi: https://doi.org/10.3390/ph16010037

2. Wang, Y., Li, D., Lv, Z., Feng, B., Li, T.,... Weng, X. (2023). Efficacy and safety of Gutong Patch compared with NSAIDs for knee osteoarthritis: A real-world multicenter, prospective cohort study in China. Pharmacological Research, 197, 106954. doi: https://doi.org/10.1016/j.phrs.2023.106954

3. Xu, X., Luo, Q., Wang, J., Song, Y., Ye, H., Zhang, X.,... Shi, G. (2024). Large-field objective lens for multi-wavelength microscopy at mesoscale and submicron resolution. Opto-Electronic Advances, 7(6), 230212. doi: 10.29026/oea.2024.230212

4. Zou, Y., Zhu, S., Kong, Y., Feng, C., Wang, R., Lei, L.,... Chen, L. (2024). Precision matters: the value of PET/CT and PET/MRI in the clinical management of cervical cancer. Strahlentherapie und Onkologie. doi: https://doi.org/10.1007/s00066-024-02294-8

5. Jia, Y., Chen, G., & Chi, H. (2024). Retinal fundus image super-resolution based on generative adversarial network guided with vascular structure prior. Scientific Reports, 14(1), 22786. doi: 10.1038/s41598-024-74186-x

6. Chen, L., Jiang, Z., Yang, L., Fang, Y., Lu, S., Akakuru, O. U.,... Wu, A. (2023). HPDA/Zn as a CREB Inhibitor for Ultrasound Imaging and Stabilization of Atherosclerosis Plaque. Chinese Journal of Chemistry, 41(2), 199-206. doi: https://doi.org/10.1002/cjoc.202200406

7. Hu, M., Yuan, X., Liu, Y., Tang, S., Miao, J., Zhou, Q.,... Chen, S. (2017). IL-1β-induced NF-κB activation down-regulates miR-506 expression to promotes osteosarcoma cell growth through JAG1. Biomedicine & Pharmacotherapy, 95, 1147-1155. doi: https://doi.org/10.1016/j.biopha.2017.08.120

8. Wang, Y., Xu, Y., Song, J., Liu, X., Liu, S., Yang, N.,... Zhang, Y. (2024). Tumor Cell-Targeting and Tumor Microenvironment–Responsive Nanoplatforms for the Multimodal Imaging-Guided Photodynamic/Photothermal/Chemodynamic Treatment of Cervical Cancer. International Journal of Nanomedicine, 19, 5837-5858. doi: 10.2147/IJN.S466042

9. Luan, S., Yu, X., Lei, S., Ma, C., Wang, X., Xue, X.,... Zhu, B. (2023). Deep learning for fast super-resolution ultrasound microvessel imaging. Physics in Medicine & Biology, 68(24), 245023. doi: 10.1088/1361-6560/ad0a5a

Comments on the Quality of English Language

The English could be improved to express the research more clearly.

Author Response

Comment 1: Concise information on multimodal approach How are CT, MRI, Bone Scan and PET/CT integrated in the multimodal approach and how does each modality uniquely contribute to the diagnosis and staging of osteosarcoma?

Response 1: Thank you for pointing this out. Our manuscript has some information in table 1 (page 3), and details of each modality, in section 2 (Imaging assessment of osteosarcoma).

 Comment 2: While the Bone-RADS classification system is based on qualitative findings, the supplementary information enhances the contextual understanding of lesions seen in practice. Expand on how it helps to stratify risk and guide the management of osteosarcoma based on imaging findings.

Response 2: Thank you for pointing this out. In table 3 (page 5) shows Bone RADS score, risk of malignancy, and recommendation.

Comment 3: Exploration of utilization of artificial intelligence and radiomics in osteosarcoma diagnosis Explain how these technologies improve the analysis and interpretation of imaging data in detail.

Response 3: We expanded the radiomics section with further references and information.

Comment 4: Explain the difficulties in early-stage diagnosis of osteosarcoma and how imaging strategies are adapted to maximize early detection.

Response 4: Thank you for pointing this out. We agree with this comment. Therefore, we have revised our manuscript in section number 4, pages 27-28.

Comment 5: Explain how MRI is used to determine the extent of a tumor and how the different MRI sequences can provide information about different tumor features.

Response 5: Thank you for pointing this out. We already mentioned the first paragraph on page 10.

Comment 6: Discuss the implications of the presence of pathological fractures on the prognosis of osteosarcoma and their influence on treatment plans.

Response 6: Thank you for pointing this out. We already added some sentences on pages 7-8 (red color).

Comment 7: They use novel imaging modalities, such as dynamic contrast-enhanced MRI and diffusion-weighted imaging, to evaluate tumor biology and treatment response.

Response 7: Thank you for pointing this out. We already added some further information and advanced MRI techniques on page 15 (red color), first paragraph (line 8).

Comment 8: Discuss the implications of the challenges of current imaging techniques for identifying skip lesions and how these factors impact surgical planning and prognosis.

Response 8: Thank you for pointing this out. We already added more sentences about skip lesions on page 10 (red color), line 16.

Comment 9: More relevant papers, especially those on advanced imaging technologies and artificial intelligence methods in bone tumor diagnosis, need to be introduced to improve the introduction and literature review.

Response 9: Thank you for pointing this out. We already added this literature.

Round 2

Reviewer 3 Report

Comments and Suggestions for Authors

Revise the paper according to my previous comments carefully and cite the relevant literature as well.

  1. Huang, H., Huang, F., Liang, X., Fu, Y., Cheng, Z., Huang, Y.,... Chen, Y. (2023). Afatinib Reverses EMT via Inhibiting CD44-Stat3 Axis to Promote Radiosensitivity in Nasopharyngeal Carcinoma. Pharmaceuticals, 16(1), 37. doi: https://doi.org/10.3390/ph16010037
  1. Wang, Y., Li, D., Lv, Z., Feng, B., Li, T.,... Weng, X. (2023). Efficacy and safety of Gutong Patch compared with NSAIDs for knee osteoarthritis: A real-world multicenter, prospective cohort study in China. Pharmacological Research, 197, 106954. doi: https://doi.org/10.1016/j.phrs.2023.106954
  1. Xu, X., Luo, Q., Wang, J., Song, Y., Ye, H., Zhang, X.,... Shi, G. (2024). Large-field objective lens for multi-wavelength microscopy at mesoscale and submicron resolution. Opto-Electronic Advances, 7(6), 230212. doi: 10.29026/oea.2024.230212
  1. Zou, Y., Zhu, S., Kong, Y., Feng, C., Wang, R., Lei, L.,... Chen, L. (2024). Precision matters: the value of PET/CT and PET/MRI in the clinical management of cervical cancer. Strahlentherapie und Onkologie. doi: https://doi.org/10.1007/s00066-024-02294-8
  1. Jia, Y., Chen, G., & Chi, H. (2024). Retinal fundus image super-resolution based on generative adversarial network guided with vascular structure prior. Scientific Reports, 14(1), 22786. doi: 10.1038/s41598-024-74186-x
  1. Chen, L., Jiang, Z., Yang, L., Fang, Y., Lu, S., Akakuru, O. U.,... Wu, A. (2023). HPDA/Zn as a CREB Inhibitor for Ultrasound Imaging and Stabilization of Atherosclerosis Plaque. Chinese Journal of Chemistry, 41(2), 199-206. doi: https://doi.org/10.1002/cjoc.202200406
  1. Hu, M., Yuan, X., Liu, Y., Tang, S., Miao, J., Zhou, Q.,... Chen, S. (2017). IL-1β-induced NF-κB activation down-regulates miR-506 expression to promotes osteosarcoma cell growth through JAG1. Biomedicine & Pharmacotherapy, 95, 1147-1155. doi: https://doi.org/10.1016/j.biopha.2017.08.120
  1. Wang, Y., Xu, Y., Song, J., Liu, X., Liu, S., Yang, N.,... Zhang, Y. (2024). Tumor Cell-Targeting and Tumor Microenvironment–Responsive Nanoplatforms for the Multimodal Imaging-Guided Photodynamic/Photothermal/Chemodynamic Treatment of Cervical Cancer. International Journal of Nanomedicine, 19, 5837-5858. doi: 10.2147/IJN.S466042

  1. Luan, S., Yu, X., Lei, S., Ma, C., Wang, X., Xue, X.,... Zhu, B. (2023). Deep learning for fast super-resolution ultrasound microvessel imaging. Physics in Medicine & Biology, 68(24), 245023. doi: 10.1088/1361-6560/ad0a5a
Comments on the Quality of English Language

 The English could be improved to more clearly express the research.

Author Response

Revise the paper according to my previous comments carefully and cite the relevant literature as well.

Article no. 1:  Huang, H., Huang, F., Liang, X., Fu, Y., Cheng, Z., Huang, Y.,... Chen, Y. (2023). Afatinib Reverses EMT via Inhibiting CD44-Stat3 Axis to Promote Radiosensitivity in Nasopharyngeal Carcinoma. Pharmaceuticals, 16(1), 37. doi: https://doi.org/10.3390/ph16010037

This article related to nasopharyngeal carcinoma. Since we wrote about osteosarcoma; we did not find anything related to osteosarcoma in this article.

Article no. 2;  Wang, Y., Li, D., Lv, Z., Feng, B., Li, T.,... Weng, X. (2023). Efficacy and safety of Gutong Patch compared with NSAIDs for knee osteoarthritis: A real-world multicenter, prospective cohort study in China. Pharmacological Research, 197, 106954. doi: https://doi.org/10.1016/j.phrs.2023.106954.

This article related to treatment of osteoarthritis. Since we wrote about osteosarcoma; we did not find anything related to osteosarcoma in this article.

Article no. 3;  Xu, X., Luo, Q., Wang, J., Song, Y., Ye, H., Zhang, X.,... Shi, G. (2024). Large-field objective lens for multi-wavelength microscopy at mesoscale and submicron resolution. Opto-Electronic Advances, 7(6), 230212. doi: 10.29026/oea.2024.230212.

This article related to multi-wavelength microscopy. Since we wrote about osteosarcoma; we did not find anything related to osteosarcoma in this article.

Article 4; Zou, Y., Zhu, S., Kong, Y., Feng, C., Wang, R., Lei, L.,... Chen, L. (2024). Precision matters: the value of PET/CT and PET/MRI in the clinical management of cervical cancer. Strahlentherapie und Onkologie. doi: https://doi.org/10.1007/s00066-024-02294-8.

This article related to PET/CT and PET/MRI of cervical cancer. It does not relate to osteosarcoma.

Article 5;  Jia, Y., Chen, G., & Chi, H. (2024). Retinal fundus image super-resolution based on generative adversarial network guided with vascular structure prior. Scientific Reports, 14(1), 22786. doi: 10.1038/s41598-024-74186-x.

This article related to images of retinal fundus. It does not relate to osteosarcoma.

Article 6; Chen, L., Jiang, Z., Yang, L., Fang, Y., Lu, S., Akakuru, O. U.,... Wu, A. (2023). HPDA/Zn as a CREB Inhibitor for Ultrasound Imaging and Stabilization of Atherosclerosis Plaque. Chinese Journal of Chemistry, 41(2), 199-206. doi: https://doi.org/10.1002/cjoc.202200406.

This article related to ultrasound imaging of arthrosclerotic plaque. It does not relate to osteosarcoma.

Article 7; Hu, M., Yuan, X., Liu, Y., Tang, S., Miao, J., Zhou, Q.,... Chen, S. (2017). IL-1β-induced NF-κB activation down-regulates miR-506 expression to promotes osteosarcoma cell growth through JAG1. Biomedicine & Pharmacotherapy, 95, 1147-1155. doi: https://doi.org/10.1016/j.biopha.2017.08.120.

This article already cited since it related to osteosarcoma.

Article no. 8; Wang, Y., Xu, Y., Song, J., Liu, X., Liu, S., Yang, N.,... Zhang, Y. (2024). Tumor Cell-Targeting and Tumor Microenvironment–Responsive Nanoplatforms for the Multimodal Imaging-Guided Photodynamic/Photothermal/Chemodynamic Treatment of Cervical Cancer. International Journal of Nanomedicine, 19, 5837-5858. doi: 10.2147/IJN.S466042.

This article related to treatment of cervical cancer. It does not relate to osteosarcoma.

Article no 9; Luan, S., Yu, X., Lei, S., Ma, C., Wang, X., Xue, X.,... Zhu, B. (2023). Deep learning for fast super-resolution ultrasound microvessel imaging. Physics in Medicine & Biology, 68(24), 245023. doi: 10.1088/1361-6560/ad0a5a.

This article related to images of ultrasound microvessels. Usually, we do not perform ultrasound in osteosarcoma. We start with plain radiographs and go to MRI. Therefore this article is rather out of scope of our review. 

Best regard,
